# A New Model of Chronic *Mycobacterium abscessus* Lung Infection in Immunocompetent Mice

**DOI:** 10.3390/ijms21186590

**Published:** 2020-09-09

**Authors:** Camilla Riva, Enrico Tortoli, Federica Cugnata, Francesca Sanvito, Antonio Esposito, Marco Rossi, Anna Colarieti, Tamara Canu, Cristina Cigana, Alessandra Bragonzi, Nicola Ivan Loré, Paolo Miotto, Daniela Maria Cirillo

**Affiliations:** 1Emerging Bacterial Pathogens Unit, Division of Immunology, Transplantation and Infectious Diseases, IRCCS San Raffaele Scientific Institute, 20132 Milan, Italy; riva.camilla@hsr.it (C.R.); tortoli.enrico@hsr.it (E.T.); rossi.marco@hsr.it (M.R.); lore.nicolaivan@hsr.it (N.I.L.); miotto.paolo@hsr.it (P.M.); 2Centre of Statistics for Biomedical Sciences (CUSSB), Vita-Salute San Raffaele University, 20132 Milan, Italy; cugnata.federica@hsr.it; 3Pathology Unit, Department of Experimental Oncology, IRCCS San Raffaele Scientific Institute, 20132 Milan, Italy; sanvito.francesca@hsr.it; 4Preclinical Imaging Facility, Experimental Imaging Center, IRCCS San Raffaele Scientific Institute, 20132 Milan, Italy; esposito.antonio@unisr.it (A.E.); colarieti.anna@hsr.it (A.C.); canu.tamara@hsr.it (T.C.); 5School of Medicine, Vita-Salute San Raffaele University, 20132 Milan, Italy; 6Infections and Cystic Fibrosis Unit, Division of Immunology, Transplantation and Infectious Diseases, IRCCS San Raffaele Scientific Institute, 20132 Milan, Italy; cristina.cigana@hsr.it (C.C.); bragonzi.alessandra@hsr.it (A.B.)

**Keywords:** *Mycobacterium abscessus*, mouse model, chronic lung infection

## Abstract

Pulmonary infections caused by *Mycobacterium abscessus* (MA) have increased over recent decades, affecting individuals with underlying pathologies such as chronic obstructive pulmonary disease, bronchiectasis and, especially, cystic fibrosis. The lack of a representative and standardized model of chronic infection in mice has limited steps forward in the field of *MA* pulmonary infection. To overcome this challenge, we refined the method of agar beads to establish *MA* chronic infection in immunocompetent mice. We evaluated bacterial count, lung pathology and markers of inflammation and we performed longitudinal studies with magnetic resonance imaging (MRI) up to three months after *MA* infection. In this model, *MA* was able to establish a persistent lung infection for up to two months and with minimal systemic spread. Lung histopathological analysis revealed granulomatous inflammation around bronchi characterized by the presence of lymphocytes, aggregates of vacuolated histiocytes and a few neutrophils, mimicking the damage observed in humans. Furthermore, *MA* lung lesions were successfully monitored for the first time by MRI. The availability of this murine model and the introduction of the successfully longitudinal monitoring of the murine lung lesions with MRI pave the way for further investigations on the impact of *MA* pathogenesis and the efficacy of novel treatments.

## 1. Introduction

Nontuberculous mycobacteria (NTMs) are environmental organisms that are ubiquitous in water and soil. The large majority of NTMs are not pathogenic to humans, causing disease only in the presence of predisposing host conditions [1]. The prevalence of pulmonary NTM infections has increased over recent decades [2], affecting individuals with preexisting lung inflammation and compromised ability to clear the infection such as in chronic obstructive pulmonary disease (COPD), bronchiectasis and, especially, cystic fibrosis (CF) disease [3]. The prevalent NTMs infecting patients with CF are members of the *Mycobacterium avium* and *Mycobacterium abscessus* (MA) complexes that include three subspecies (subsp.), *MA* subsp*. abscessus*, *MA* subsp. *bolletii* and *MA* subsp. *massiliense* [4]. MA is a rapidly growing NTM responsible for chronic pulmonary infections associated with diversified clinical presentations ranging from asymptomatic colonization to a significant decline of lung function associated with morbidity and mortality, and with poor treatment outcome. Despite prolonged therapy with multiple antibiotics used in different combinations, *MA* infections are extremely difficult to eradicate due to the multidrug resistance of the mycobacterium [5,6], which restricts the number of molecules available for the treatment. Among the multiple cellular and animal models proposed for studying the pathology of *MA* infection, very few could be used to evaluate new drugs’ efficacy [7]. Murine and human primary macrophages or other cell lines were used to dissect the early *MA* invasion of phagocytic cells. Amoebae were useful as host models to study and identify the expression of several determinants that contribute to *MA* virulence [7]. Zebrafish (*Danio rerio*) embryos were exploited during the last two decades to study the intricate interactions between *MA* and the host immune system. Immunocompetent mouse models of infection following aerosol [8], intratracheal [9], or intravenous [10] bacteria inoculation, were characterized by transient colonization with a rapid clearance of *MA* in the first weeks post challenge. Alternative models based on immunocompromised murine models were developed (e.g., granulocyte-macrophage colony-stimulating factor GM-CSF -/- mice, gamma interferon knockout GKO mice or severe combined immunodeficiency SCID mice) [11,12,13], which have resulted in slower resolution of *MA* lung infection than immunocompetent mice. Despite these advancements, the lack of a standardized mouse model of persistent chronic infection in immunocompetent mice that reflects the human *MA* lung pathology, is the major impediment to studying *MA* pulmonary infection and treatment strategies [14]. In this study, we used *MA* subsp*. abscessus*, *MA* subsp. *bolletii* and *MA* subsp. *massiliense* to establish a long-term chronic pulmonary infection in immunocompetent mice. To infect mice, we adapted to *MA* a method based on intratracheal inoculation of bacteria embedding agar beads, previously used with other pathogens (*Pseudomonas aeruginosa*, *Staphylococcus aureus* and *Burkholderia cenocepacia*) [15,16,17]. We monitored and analyzed the host response (bacterial burden, pulmonary lesions and inflammatory cytokines) for up to 90 days from infection. Furthermore, we explored the possibility of longitudinally monitoring the *MA* lung challenge within the same animal by using the MRItechnique for the first time.

## 2. Results

### 2.1. Chronic Lung Persistence of MA in C57BL/6NCrl Mice

MA subsp. *abscessus* embedded in the agar beads was intratracheally inoculated in immunocompetent C57BL/6NCrl mice to generate an effective long-term chronic lung infection. Different groups of mice were sacrificed at different time points (7, 14, 28, 45, 65, 90 days). Results showed that up to 65 days after infection, the overall rate of *MA* subsp. *abscessus* chronicity was 97.56% (Figure 1A). The colony forming units (CFUs) were not significantly different among the time points 7, 14, 28, 45 and 65 (Kruskal–Wallis test *p*-value = 0.0752) with a stable bacterial load (median: ~10^6^ CFUs) (Figure 1B). After this time point, some mice started clearing the bacteria and the lung chronicity rate significantly decreased to 40% (Figure 1A, Appendix A). *MA bolletii* (Appendix A) and *MA massiliense* (Appendix A) were chronically infected for up to 45 days, but after 7 days of infection the number of CFUs started to decrease with statistical significance. Of note, lung infection of the three subspecies led to a minimal systemic spread as indicated by the number of CFU observed in the spleen (Appendix A). The body weight (Appendix A) and the Kaplan–Meier survival curves (Appendix A) of the three subspecies did not show differences in comparison with control mice, injected with empty beads.

### 2.2. Lung Lesions and Inflammatory Response during MA Chronic Infection in C57BL/6NCrl Mice

Histopathological characterization of lungs infected by *MA* subsp. *abscesuss* was performed at different time points for 3 months. Morphological analysis after 7 days of infection revealed aggregates of inflammatory cell infiltrate around bronchi involving surrounding parenchyma (Figure 2A), characterized by IBA-1 positive-histiocytes, some of which were vacuolated histiocytes, and a few CD3 and B220 positive lymphocytes (Figure 2A and Appendix A). After 45 days of infection, the inflammatory cell infiltrate was organized as granuloma-like structures mainly localized in the parenchyma around bronchi (Figure 2A) and characterized by a central core of IBA-1 positive-histiocytes intermingled with a few neutrophils (Figure 2A), surrounded by, CD3 and B220 positive- lymphocytes (Appendix A). Microscopic analysis after 90 days showed a peribronchial cuffing (Figure 2A) of CD3, B220, and IBA-1-positive cells (Appendix A). *MA* subsp. *abscessus* lung localization after 45 days of infection was detected by Ziehl Neelsen staining showing the bacteria intermingled with beads forming a plug, surrounded by neutrophils, located in the lumen of the main bronchi (Appendix A). The quantification of the total lung lesions (inflammatory cell infiltrate, granuloma-like lesions and bleeding area) induced by the *MA* subsp. *abscessus* was statistically higher than in control mice, during the course of the infection (Figure 2B,C, Appendix A). Empty beads in control mice-induced mild inflammatory cells were characterized by infiltration by lymphocytes, plasmacells and a few histiocytes but these lung lesions are less severe and show a different morphology compared to the granuloma-like structures that we observed in mice infected with MA. Similar results were obtained with *MA* subsp. *bolletii* and *MA* subsp. *massiliense* (Appendix A). Tumor Necrosis Factor (TNF-α) (Figure 3A) and Interferon-gamma (IFN-γ) (Figure 3B) showed a significant peak expression at an early time point of *MA* subsp *abscessus* infection compared to control mice. These results were confirmed for *MA* subsp. *bolletii* (Appendix A) and *MA* subsp. *massiliense* (Appendix A).

Longitudinal monitoring by MRI was introduced to evaluate the progression of the pulmonary lesions within single mice at the day of infection and at the time points of 7, 45, 90 days. Lesions induced by *MA* subsp. *abscessus* were clearly visible at the time points of 45 and 90 days. In particular, infection foci appeared as a round well-defined area of consolidation with a preferential distribution in the left lung (Figure 4).

## 3. Discussion

The emerging problem of *MA* infections in the last 20 years has focused attention on the understanding of the pathogenesis of *MA* disease, which remains inadequately characterized [18]. Our aim was to develop an immunocompetent mouse model overcoming *MA* clearance and miming human *MA* airway disease (e.g., CF, COPD, bronchiectasis or asthma). We performed intratracheal infection in C57BL/6NCrl mice, known as resistant to mycobacterial infection, with *MA* entrapped in small agar beads. This method was refined from the original protocol established by Bragonzi et al. successfully used to study *P. aeruginosa, B. cepacia* and *S. aureus* infection and co-infections [15,16,17]. So far, C57BL/6NCrl mice were characterized by efficient clearance of *MA* from the lungs when inoculated via aerosol at a low dose; in contrast, airways clearance was much less effective following inoculation of high intravenous doses, ranging from 10^6^ to 10^8^ CFU [19,20]. Furthermore, male C57BL/6NCrl mice are described as developing more severe lung lesions and having greater mortality with mycobacterial infection than female C57BL/6NCrl mice [21]. Embedding *MA* within agar beads and using intratracheal injection physically retained the bacteria in the bronchial airways and provided microanerobic/anaerobic conditions that allow bacteria to grow in microcolonies [22]. This procedure allowed us to create a stable infection in immunocompetent male mice with all the three *MA* subspecies, and to reproduce a chronic infection useful for studying pathogenesis and progression of *MA* lung disease until three months from the inoculum. In particular, the chronicity breakpoint was, for *MA* subsp*. abscessus,* 65 days and, for *MA* subsp*. bolletii* and *MA* subsp*. massiliense*, 45 days; after such time points, mice progressively inclined to clear the infection. Very recently, *Le Moigne* [23] have published that intratracheal infection with agar beads is associated with very high mortality, in our hands the mortality has been very low and the chronicity was successfully established. In human hosts, *MA* can persist silently for years, and even decades, and can induce organized granulomatous lesions with foamy cells and in some cases, even caseous necrosis, like *M. tuberculosis* [24]. Infection with intracellular pathogens like *MA* is effectively controlled by the cell-mediated immune response and several cytokines play important roles limiting the host inflammatory response. In particular, IFN-γ is critical in the immune response to *MA* infection: it activates infected macrophages in concert with TNF-α and initiates a major effector mechanism of cell-mediated immunity. Immunocompetent mice, with evidence of strong TH1 response [25], appear resistant since they are characterized by a transient lung infection with a rapid clearance of *MA* in the first weeks [8,9,10]. Several attempts, with the aim of slowing down the resolution of *MA* lung infection, have led to establishing mouse models with deficits in innate or acquired immunity (e.g., GM-CSF -/, GKO or SCID mice) [11,12,13]. Moreover, corticosteroids were recently administered to immunocompetent mice to increase their susceptibility to the aereosolized pulmonary *MA* infection [26]. In our murine model, despite an initial protective high release of TNF-α and IFN-γ, granuloma-like lung lesions (characterized by aggregates of histiocytes, a few neutrophils, and lymphocytes) were present around bronchi involving surrounding parenchyma for about two months after *MA* infection, likely mimicking the damage observed in humans [24]. Interestingly, the presence of vaculated histiocytes, reported previously only in immunocompromised mice (SCID mice and GM-CSF -/- mice) [12], highly supports the reliability of this model with immunocompetent mice. For the first time, we introduced the monitoring of the *MA* lung infection with the MRI technique in this study. MRI represents a non-invasive approach providing an excellent qualitative evaluation for the assessment of inflammatory soft-tissue diseases in both humans and small laboratory animals [27], as well as an effective tool for monitoring lung inflammatory processes [28], without the possible detrimental impact on the immune system related to the use of X-ray-based imaging approaches like Computed Tomography. MRI was applied in longitudinal measurements to monitor changes in infectious lesions, within the same C57BL/6NCrl mice, at the day of infection and after 7, 45, and 90 days from infection. Despite the decrease in pro-inflammatory markers and the dissolutions of granuloma-like lesions during the course of the infection, the murine lung parenchyma was still damaged at 90 days, as observed following both histopathological and MRI analysis, indicating a correlation between the two techniques. MRI was also able to discriminate, as with the histopathological analysis, the lung inflammatory foci, induced by empty beads in control mice presenting as microscopic areas of consolidation at day 45 and 90.

MA is probably the most pathogenic of the NTMs infecting patients with CF due to its multidrug resistance, poor response to treatment, and association with a decline in lung function [29,30,31]. In patients with CF, *MA* pulmonary infections are associated with a wide clinical spectrum of disease, ranging from asymptomatic, transient colonization to significant lung function decline; making problematic the decision to start a combination therapy often associated with significant toxicity [31,32,33]. Published data show that the lack of suitable models of chronic infection jeopardise the urgent need to study the pathogenesis and the host immune response of *MA* lung disease. We show, for the first time, a long-term chronic infection model for all the three *MA* subspecies in immunocompetent male C57BL/6NCrl mice, with similar clinical manifestations of the disease among the three subspecies as observed in humans [34]. We investigated the *MA* infection for up to six months when very few mice were still infected (data not shown) and the lung parenchyma was characterized by a residual peribronchial cuffing of CD3, B220, and IBA-1-positive cells (data not shown). Furthermore, we introduced the MRI technique for the first time in order to monitor the *MA* subspecies chronic lung infection over time by correlating imaging results with microbiological, immunological, and histological analyses. We acknowledge that our model has several limitations. Experiments were conducted only in male C57Bl/6NCrl mice, described as more susceptible to mycobacterial infection than female C57BL/6NCrl mice [21] and with reference strains only. Infection of other genetic backgrounds, with clinical strains or strains with a stable, rough virulent phenotype [35], could have provided different results. Our ongoing preliminary experiments (data not shown) indicate that chronicity could be established in CF mice and extended for longer than 90 days. We have shown that it is possible to monitor lesion progression in a single animal at different times by MRI. This makes it possible to decrease the number of animals included in studies of this kind. We noticed that the lesions induced by empty beads can be differentiated from the granuloma-like structures induced by beads containing MA. Despite these critical points, our *MA* murine model could be adapted to study strain-related virulence in CF and non-CF populations; representing a new suitable preclinical model for evaluating new antibiotics and innovative regimens.

## 4. Materials and Methods

### 4.1. Ethics Statement

Animal studies were approved by the Italian Ministry of Health guidelines for the use and care of experimental animals (3 November 2016) and registered by San Raffaele Scientific Institute Institutional Animal Care and Use Committee (IACUC N°816, 11 November 2016).

### 4.2. Bacterial Strains

MA subsp. *abscessus* (ATCC 19977) [36], *MA* subsp. *bolletii* (ATCC 8156) [37] and *MA* subsp. *massiliense* (ATCC 48898) [38] were used. All the strains were conserved frozen at −80 °C and were thawed and grown on Middlebrook 7H10 plates for three days at 37 °C.

### 4.3. Mouse Strains

Immunocompetent C57BL/6NCrl male mice (8 to 10 weeks of age) from Charles River were tested in the experiments. All mice were maintained in specific pathogen-free conditions at San Raffaele Scientific Institute, Milan, Italy.

### 4.4. Mouse Model of Chronic MA Lung Infection

Some colonies from 7H10 plates were grown for 2 days (to reach the exponential phase) in 20 mL of Middlebrook 7H9 broth. Then bacteria were embedded in agar bead preparation with modifications to the standard protocol [39]. The preparation of the beads was particularly challenging and required six months to achieve satisfactory work. Key factors included the amount of white heavy mineral oil that was reduced (50 mL) and the amount of Trypticase Soy Agar (TSA) that was increased (25 mL). Ten accurate washing steps were needed to remove traces of the mineral oil. On the day of the agar bead preparation, 5 or 6 agar bead batches were made. The stirrer was set at medium speed to mix bacteria, oil and agar with a magnet to generate the correct size of the agar beads (100–200 μm). The success of the agar beads preparation was ~50%. Agar bead preparations were stored at 4 degrees for no more than a week. After this time, the number of the viable bacteria included in the agar beads decreased. The preparation was performed fresh every time.

For the inoculum, mice were anesthetized, the trachea was exposed and intubated, and 50 µL of beads suspension (1 × 10^5^ CFU) were injected before closing the incision with suture clips. Control mice were intratracheally inoculated with the same volume of empty beads suspension. The best agar bead size to intratracheally infect mice and to establish a chronic infection is 100–200 μm. Agar bead sizes smaller than 100–200 μm could be easily cleared by mice, while sizes larger than 100–200 μm may not reach the lung during the intratracheal infection. After infection, mice were daily monitored for body weight, appetite and hair coat; at fixed time points from infection (7, 14, 28, 45, 65 90 and 180 days). On average, six mice were euthanized by CO_2_ asphyxiation.

Lung and spleen were collected and, once homogenized, were processed for microbiological analysis. Total CFU were the result of the addition of the CFU in lung homogenate and bronchoalveolar lavage fluid (BALF). The lung supernatants were stored at −80 °C for cytokines and chemokines determination. In particular, IFN-γ and TNF-α were evaluated by Mouse Custom ProcartaPlex 9-plex (Invitrogen, Thermo Fisher Scientific, Waltham, MA, USA) and normalized at 2500 ug/mL of quantified proteins in lung supernatants.

### 4.5. Histological Analysis and Lung Damage Quantification

Formalin-fixed, paraffin-embedded sections of lungs at 7, 45 and 90 days from *MA* infection were stained with H&E for histopathologic analysis. Immunohistochemical staining was performed with rabbit polyclonal anti-IBA-1 (Wako), monoclonal rat anti-human CD3 (Bio-Rad, Berkeley, CA, USA) and rat anti-mouse B220/CD45R (clone RA3-6B2; Bio-Rad), after antigen retrieval. Immunoreactions were revealed by rabbit or rat on rodent HRP-polymer kit (Biocare Medical, Columbus, OH, USA), using 3,3 diaminobenzidine (DAB) as chromogen (Biocare Medical) and slides were counterstained with haematoxylin. Ziehl–Neelsen staining was performed using an automated Benchmark Special Stains instrument and dedicated staining kit (Roche, Basel, Switzerland). Photos of immunohistochemical staining and Ziehl–Neelsen staining were taken using AxioCam HRc (Zeiss, Jena, Germany) with the AxioVision System SE64 (Zeiss).

Lung tissue lesions were quantified by Fiji ImageJ software [40] (version 1.52p) in 5 lobe images (Magnification 5×) of 5 mice for each time point (7, 45 and 90 days from *MA* infection). In particular, the analysis was performed by Waikato Environment for Knowledge Analysis (WEKA), release v3.2.33 [41] (available at https://imagej.net/Trainable_Weka_Segmentation). The algorithm was trained to distinguish vessels/bronchi, lesions, healthy tissue, beads, blood, empty beads, artifacts/shadow (training set: at least 10 selections for each of the seven classes listed). Analysis was reported as the percentage of lung lobe lesions to the percentage of lung lobe area.

### 4.6. MRI Longitudinal Monitoring

MRI studies were performed on a 7T preclinical scanner (Bruker, BioSpec 70/30 USR, Paravision 5.1, Ettlingen, Germany), equipped with 450/675 mT/m gradients (slew-rate: 3400–4500T/m/s; rise-time 140 μs) and a circular polarized mouse body volume coil with an inner diameter of 40 mm. MRI acquisitions were performed during gas anesthesia (Isofluorane, 3% for induction and 2% for maintenance 2 L/min oxygen) and physiological monitoring. MRI scans were conducted with mice placed in a prone position inside the animal bed, scanning at fixed time points (7, 45 and 90 days from *MA* infection). MRI protocol included: Three-Dimensional Intra-Gate Fast Low Angle Shot (3D IG FLASH) sequences acquired in the axial and coronal planes and a Two-Dimensional Ultra short Echo Time (2D UTE) sequences in the axial plane. Sequence parameters were TR = 10 ms, TE = 2 ms, FOV = 25 × 24 mm, spatial resolution = 0.098 × 0.078 × 0.469 mm/pixel for 3D IG FLASH and TR = 30 ms, TE = 0.478 ms, FOV 26 × 26 mm, spatial resolution = 0.117 × 0.117 mm/pixel, slice thickness 0.9 mm for 2D UTE.

### 4.7. Image Analysis

Following acquisition, lung images were transferred to a dedicated workstation and analyzed with the use of MIPAV (Medical Image Processing, Analysis, and Visualization, National Institute of Health, Center for Information Technology). Images were analyzed by experienced radiologists. A region-of-interest (ROI) was drawn for each lung lesion on each slice on 3D-IG-FLASH axial images; the presence of the lesion was also confirmed by comparing coronal 3D-IG-FLASH and axial 2D UTE images, to avoid inclusion of potential artefacts. After segmentation of 3D-IG-FLASH axial images, the whole volume of the affected lung parenchyma was automatically computed.

### 4.8. Statistics

A linear mixed-effects (LME) [42] model was employed to estimate the longitudinal trend of the body weight (log10 transformation) and evaluate the differences among groups (Control, *MA* subsp. *abscessus*, *MA* subsp. *bolletii* and *MA* subsp. *massiliense*). Different trends were allowed for Days ≤ 3 and Days > 3. For Days > 3, both linear and quadratic terms for time were included in the mixed models to account for the nonlinear trajectories of body weight over time. LME models were also employed to estimate the longitudinal trend of the bacterial load in the spleen and in the lung and the tissue damage. The *Ordered Quantile* normalization transformation of the outcome was considered in order to satisfy underlying model assumptions. Post-hoc analysis after LME was performed considering all the pairwise comparisons.

Logistic regression models were used to evaluate the lung chronicity rate during the course of the infection. The overall survival curves were estimated with the Kaplan–Meier method and were compared using the log-rank test. The Mann–Whitney test was performed to compare two independent groups, while in the presence of more than two independent groups, the Kruskal–Wallis test, followed by post-hoc analysis using Dunn’s test, was used. Analyses were performed using R statistical software.

## Figures and Tables

**Figure 1 ijms-21-06590-f001:**
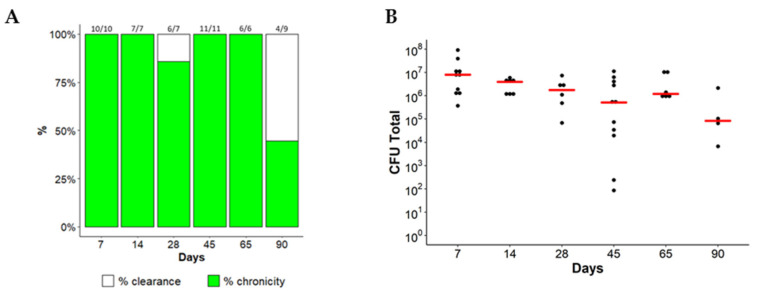
Time course of *MA* subsp. *abscessus* chronic lung infection. C57BL/6NCrl mice were intratracheally infected with 1 × 10^5^ of MA *abscessus* reference strain (ATCC 19977) embedded in agar-beads. After 7, 14, 28, 65, 90 days from infection, mice were sacrificed. (**A**) Rate of chronicity (number of mice = 6–11). The data were pooled from two independent experiments (Appendix A). (**B**) Mice were considered infected with CFUs > 0 in total lung. Dots represent CFUs in individual mice. The red line represents the median values. The Kruskal–Wallis test was performed.

**Figure 2 ijms-21-06590-f002:**
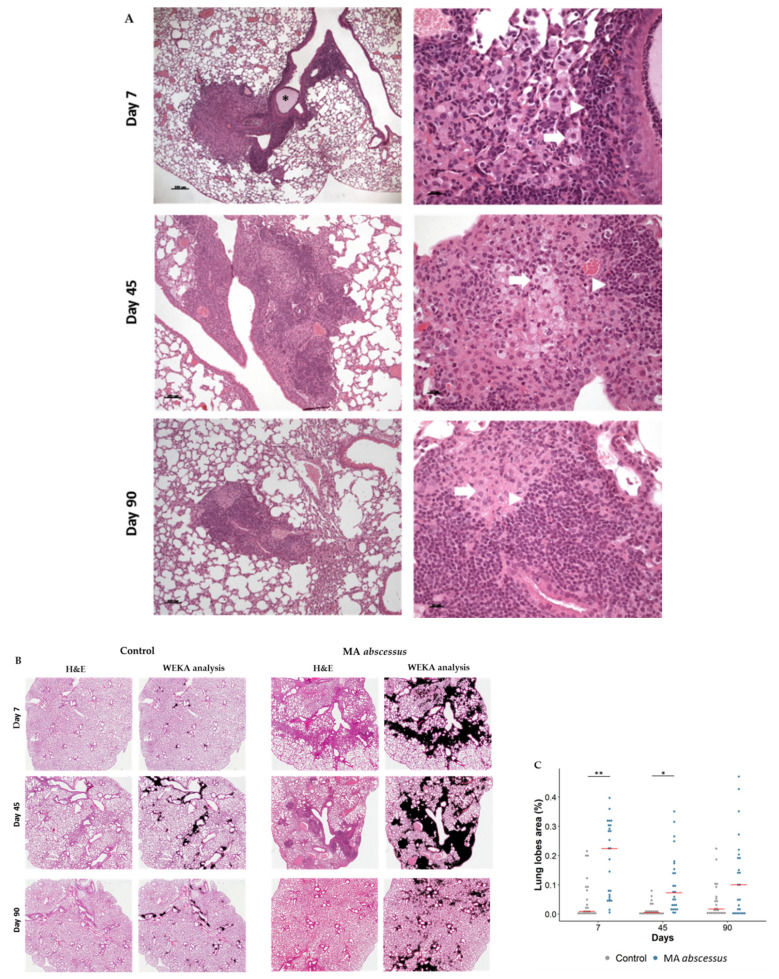
Lung lesions after 7, 45 and 90 days of chronic infection with *MA* subsp. *abscessus*. (**A**) The panel shows H&E, (10× on the left and 40× on the right, AxioCam HRc Zeiss)-stained sections of lungs after 7, 45 and 90 days from infection; scale bar: 100 μm and 20 μm. Asterisk indicate beads in the bronchial lumen, arrows indicate vacuolated histiocytes and triangles indicate lymphocytes. (**B**) The panel shows H&E and WEKA segmented images (Magnification 5×) of control and infected mice (**C**). Dots represents the lobes analyzed by the Image J program with the WEKA algorithm: 5 lobe images of 5 mice for each time point (25 dots for each time point). Total lung lesions were estimated as the percentage of lesioned area to total lung area in mice infected or not by *MA* subsp. *abscessus* (line at median). A linear mixed model followed by post hoc analysis was performed (Appendix A). Significance was calculated for each time point, comparing *MA* subsp. *abscessus* and control. * *p* < 0.05, ** *p* < 0.01.

**Figure 3 ijms-21-06590-f003:**
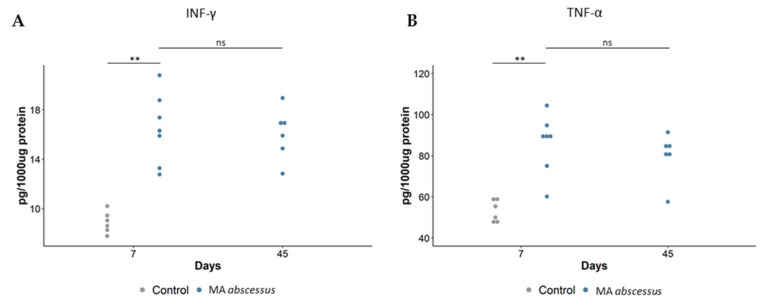
**Cytokines/chemokines recruitment during *MA*****subsp.*****abscessus*****chronic lung infection.** (**A**) IFN-γ, (**B**) TNF-α, measured by Mouse Milliplex, were quantified in total lung of the mice. At 7 and 45 days, dots represent cells in individual mice selected from the group of infected mice with *MA* subsp. *abscessus*. The data were pooled from two independent experiments. For *MA* subsp. *abscessus*, statistical time effect was evaluated with a Mann–Whitney Test. Statistical comparison between *MA* subsp. *abscessus* and control mice at Day 7 was calculated with a Mann–Whitney Test. ** *p* < 0.01.

**Figure 4 ijms-21-06590-f004:**
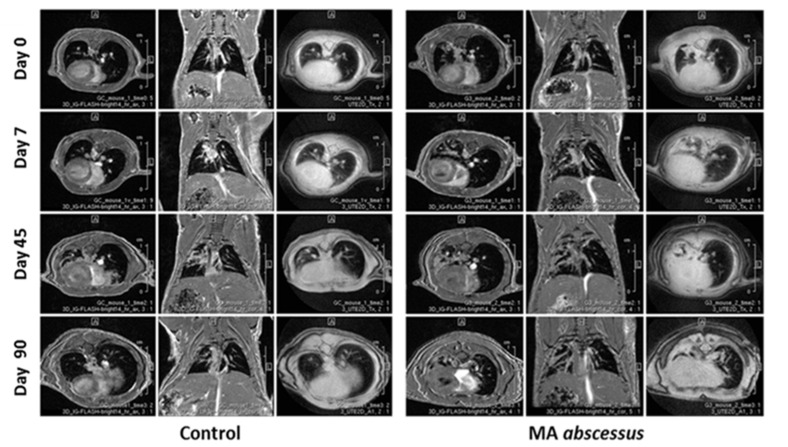
MRI analysis during the course of chronic *MA* subsp *abscessus* infection. Examples of representative axial and coronal 3D-FLASH and axial (2D-UTE) MR images of single mice lungs obtained after inoculation with sterile agar beads (control mice) and with *MA* subsp. *abscessus*. The two mice (control mice and infected mice) monitored by MRI over time were always the same from Day 0 to Day 90.

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
