# Peer review of "A New Model of Chronic Mycobacterium abscessus Lung Infection in Immunocompetent Mice"

_ijms, 2020, doi:10.3390/ijms21186590_

Round 1

Reviewer 1 Report

Currently, there are not many animal models for M. abscessus infection that mimic human chronic M. abscessus infection.

The authors discovered a very effective animal model using M. abscessus embedded in the agar beads and immunocompetent mouse named C57BL/6NCrl. 

In this model, M. abscessus clearance was not observed until 90 dpi and observed total lung lesions. Furthermore, MRI analysis supports the above-described observation.

Question: M. abscessus infection can be divided into two different morphotypes. One is smooth and the other one is rough. Normally, R morphotype shows much hypervirulence than S morphotype. In vivo efficacy study, if we use R morphotype for infection we will observe significant number of fatal mouse within short days in immunocompetent mice.

Is there also a clearance problem in vivo efficacy with R-morphotype in an immunocompetent mouse?

Thus, I suggest that the author should describe the current in vivo efficacy animal model with R-morphotype as well. 

Reviewer 2 Report

This manuscript provides a potentially very important advance, describing a relatively sustained model of Mycobacterium abscessus in immunocompetent mice. This has proved difficult to achieve in the past, and the authors appear to have made the breakthrough using agar beads carrying M. abscessus. The authors also provide a potentially important advance in describing serial imaging of pulmonary lesions through magnetic resonance imaging. If these advances can be readily validated and used in other labs, they unquestionably have the potential to move the field forward. The main parts of the manuscript are well-written, and the existing literature is well summarised. 

Because this is largely a technical paper, several criticisms can be made, and clarity could be provided in a number of places to assist researchers seeking to adopt the model. However, these do not significantly detract from the central potential importance of the paper. 

MAJOR COMMENTS

1. The agar-bacterium mix is produced in-house, and seems likely to be subject to significant variation, from operator to operator, and presumably from lab to lab. The authors' labs have clearly worked on this technique for many years, with this and other pathogens, and it is unclear how readily transferable their techniques are. No indication is given of how many times the preparation of agar-bacterium fails from batch to batch. It is unclear whether agar beads can be "stored" for later use (presumably they cannot, but if they can, it would be excellent if the authors would make these available, rather than new labs having to make fresh every time, not least from a quality control perspective). It is not clear whether they tried commercial agar beads (presumably they did, and this did not yield good results). It is not clear why (based on their JoVE 2014 paper using Pseudomonas), the "bead diameter must be between 100-200 microns" - what if it is not, and how many times is it not? The sentence on line 231 "A lot of accurate washing steps were needed to remove traces of the mineral oil." does not allow independent researchers to know exactly what to do to get over what sounds like a challenging and crucial step. It is not clear how many months/years it took them to perfect the technique. In other words, the usefulness and importance of this paper lies in how easily it can be transferred to other interested labs. The paper would have been stronger if this had been validated in another group already, but in the absence of such validation, the methodological challenges should be much more clearly defined.

2. The histology, and its interpretation, let this very good paper down. Figure 2 and Figure S5 are particularly poor. The authors frequently refer to the peribronchial, parenchymal distribution of lesions, yet they only show medium power views (or rather, very small micrographs from high power views). What would help the reader much more would be high-quality low power views (so that the distribution of lesions in the whole lung can be seen, and their relationship to bronchi truly assessed) and high-quality high power views, so that cell types can be discerned (and pointed out). The authors describe the importance of "granulocytes" and "foamy histiocytes" in the pulmonary lesions, but this is impossible for the reader to verify at this magnification. By granulocytes, presumably they mean neutrophils (I can't readily see eosinophils) - they do not use neutrophil or eosinophil stains in their immunohistochemistry, and perhaps they should in a revision? They have not picked out foamy macrophages at high power, and demonstration of multinucleate giant cells (if they found any) would be most useful. Reference to vacuolated foamy cells and neutrophils in the abstract is inappropriate at present. Figure S5 is almost impossible to interpret, when the low power view is not shown, and we are not told how often such lesions are found, although I concede that the ZN staining is extremely useful.

3. After an excellent introduction, methods and results, the discussion is weak. This is because no consideration is given to the limitations. What the reader (and specifically interested labs) really want to know is "how representative of human disease is this, and how difficult is this to do?" In my view the paper would be significantly improved by a frank assessment of limitations. The agar itself appears to me to be pathogenic, and the authors need to address the contribution from the agar beads (as opposed to the M. abscessus itself). Potential limitations of the method should be addressed. The model is an improvement, but ultimately is still not completely "chronic and established", and this should be addressed. A key factor is that the authors have not shown an effective therapeutic effect in this paper and this should be acknowledged (particularly if it was attempted). Another major issue is that they have not extended their findings to a diseased immunocompetent mouse - this is not a criticism per se (and presumably that may be their next step), but they should acknowledge that doing this in a CF or COPD mouse, while immensely challenging, would be still more useful; frustratingly they do describe CF mice in their methods (line 253), but I presume this is a typo rather than something they have actually done? Why only macrophage, pan-T cell and B cell markers were used should be discussed. The differences between this and human disease should be discussed. More points like these could be added, but the point is that, instead of a slightly self-congratulatory discussion, the paper would be far stronger by also having a careful reflection of why the model is not perfect yet.

MINOR COMMENTS

4. The conclusion that systemic infection is limited is over-interpreted. As far as I can tell, 3 of 4 mice still infected with M. abscessus at 90 days had splenic infection. This should be toned down. 

5. The use of MR is impressive. However, are we shown the same mice, imaged serially, in Fig 4? This is not clear. There are significant limitations to "standard" MR imaging of the lung. Also, the presence of disease in the control mouse in a similar distribution (even presumably using their "best pictures for publication"), again raises the question of the effect of the agar. These issues perhaps merit discussion in the limitations section?

6. In this field the background of the mice used seems to be critical. The background of the C57BL/6NCrl mice used should be more explicitly discussed, along with strengths and weaknesses. Similarly, they should justify why only male mice were used. Given the known differences in immunity in female and male mice, it is a pity that they did not extend their findings to female mice, and this should be discussed. 

7. Did they use smooth or rough colonies? Where were the ATCC strains derived from (human pathology?)?

8. Were the researchers blinded when making their observations? Presumably not? This should be acknowledged, particularly given the apparent disease in agar control mice. 

9. I am no statistician, but I found the statistical tables in the Supplement to be esoteric and hard to interpret. For me, these distracted away from the most relevant information. The authors infer that these give information about "break points for chronicity" and relevance (or not) of systemic infection. Either this should be much more clearly explained, or a statistician should look at the manuscript to check for accuracy of interpretation, or some of this might be removed?

10. Fig 2B is poorly explained. What does the y-axis indicate? When each dot represents a score in one lobe from one mouse, and when the median (to my eye) at d45 in the infected mice is <0.1, it is impossible to work out what <0.1 of a lung lesion means. This does suggest that the number of lesions in the lung may be very low (this is compounded by the absence of low power views, etc).

11. Lines 154-6 imply that the beads allowed bacteria to be physically trapped "in airways". What level of airway? There are no data presented to support this sentence. 

12. The description of antibodies used (around line 248) is confusing. For the primary antibodies, why did they use an anti-human antibody for CD3? Surely all primaries should be raised against the relevant murine antigen? The secondary antibodies are more confusing, with the statement that they used "rabbit or rat on rodent HRP-polymer". Presumably they used an HRP-conjugated antibody against the species of antibody used in the primary? This should be much clearer. 

13. In general, "CF patients" should be changed to "people with CF" or "patients with CF".

14. The references should be tidied up a little. Some of the references do not have full page numbers. Reference 34 (key to the methodology), should be cited as it is in PubMed. 
